# Effect of external therapies of traditional Chinese medicine on constipation in patients with CRF: A meta-analysis

Yu Wu[1], Qisu Ying[1], Yajing He[1], Xiangcheng Xie[1], Xiao Yuan[2], Ming Wang[1], Xiao Fei[1], Xiu Yang[1]*

1 Department of Nephrology, Affiliated Hangzhou First People's Hospital, Zhejiang University School of Medicine, Hangzhou, China, 2 Department of Nursing, Affiliated Hangzhou First People's Hospital, Zhejiang University School of Medicine, Hangzhou, China

* yxyyxx@126.com

## Abstract

### Objective

The purpose of this study was to evaluate the curative effect of external therapies of traditional Chinese medicine on constipation in patients with chronic renal failure and to provide scientific theoretical basis for clinical practice.

### Method

We searched the English database of PubMed, EMBASE, the Cochrane Library and the Web of Science and Chinese database of CNKI, Wan fang database, VIP Database and China Biomedical Literature Database up to December 2022. Randomized controlled trials (RCTs) involving constipation in patients with CRF that compared external therapies of traditional Chinese medicine and routine treatment to routine treatment were eligible for the analysis. A meta-analysis of the outcome indicators including total efficiency, weekly defecation times, defecation time, defecation difficulty score, patient-assessment of constipation quality of life and adverse events of treatment were performed. The analysis was performed by using Review Manager version 5.3.

### Result

A total of 23 studies were included, with 1764 patients. Meta-analysis results showed that compared with the control group, the test group could significantly increase weekly defecation times(MD = 0.94, 95%CI(0.70, 1.18), Z = 7.74, P < 0.00001), reduce defecation time (MD = -2.92, 95%CI(-3.69, -2.16), Z = 7.49, P < 0.00001), reduce defecation difficulty score (MD = -1.92, 95%CI(-2.45, -1.39), Z = 7.11, P < 0.00001), improve the quality of life in patients with constipation(MD = -7.57, 95%CI(-10.23, -4.91), Z = 5.58, P < 0.00001) and obtain a higher total effective rate of treatment(OR = 4.53, 95%CI(3.27, 6.29), Z = 9.07, P < 0.00001). In terms of safety, there was no statistical significance in the incidence of adverse events between two groups(OR = 0.35, 95%CI(0.04, 2.95), Z = 0.96, P = 0.34).

**Data Availability Statement:** All relevant data are within the manuscript and its Supporting information files.

**Funding:** This study was supported by Grants from Zhejiang Provincial Traditional Chinese Medicine Science and Technology Project (2021ZB223). The funders had no role in study design, data collection and analysis, decision to publish, or preparation of the manuscript.

## Conclusion

The combination of external therapies of traditional Chinese medicine and routine treatment could achieve an excellent curative effect, and there was no specific adverse event. However because of the limited level of current evidence, more high-quality trials are needed in the future.5

## 1. Introduction

Constipation is a common disease, It is reported that constipation is a symptom that affects 11–20% adult population every year [1]. And the mean prevalence rate of chronic constipation is approximately 14% all over the world [2]. In China the prevalence rate in adults is 7.0% ~20.3% [3]. From the most recently published diagnosis criteria which is from Rome Committee in 2016 (Rome IV), The definition of constipation includes that straining, lumpy or hard stools, the sensation of incomplete evacuation, the sensation of anorectal obstruction or blockage, the use of manual manoeuvres to facilitate defecation, and fewer than three bowel movements per week, any two or more items of these symptoms continued for at least 3 months in the preceding 6 months can qualify the patient for a diagnosis of functional constipation [4]. Constipation may occur because of a primary motor disorder involving the colon or can be induced by secondary causes, It may be associated with a large number of diseases or as an adverse effect of many drugs [5]. Chronic renal failure (CRF) is the outcome of the continuous progression of various chronic kidney diseases, with metabolite retention and imbalance in the water, electrolytes, and acid base being the main manifestations of CRF [6]. For example, uremia in CRF patients, electrolyte disorders like hypercalcemia, hypokalemia (severe) and hypothyroidism (severe) which are common symptoms in uremia are some of important causes of constipation [5]. And constipation is also a more common problem for patients receiving hemodialysis [7]. According to statistics, approximately 53% patients with end-stage renal disease who are receiving dialysis are suffering from constipation [8]. And the incidence of constipation is 71.7% in haemodialysis patients and 14.2% in peritoneal dialysis patients respectively [9]. A global multicenter study showed that chronic constipation could significantly reduce the quality of life, as well as in China [10, 11]. The treatment of constipation includes general treatment of reasonable diet, plenty of water, moderate exercise and establishing good defecation habits, which are the basic treatment measures for patients with chronic constipation, and medical treatment includes the administration of stimulant and osmotic laxatives, new intestinal secretagogues and peripherally restricted μ-opiate receptor antagonists et al. [5, 12]. Abusing of laxatives or repeated medical treatment can cause patients a heavy financial burden [13]. For the past few years, external therapies of traditional Chinese medicine had played a very important role in the treatment for constipation in people diagnosed with chronic renal failure [14].

There are various ways of external therapies of traditional Chinese medicine of constipation, including acupuncture, moxibustion, massage, cupping therapy and so on. A large number of clinical studies had shown that the external therapies of traditional Chinese medicine were not only effective, but also could it avoid some adverse consequences such as abdominal pain, electrolyte disturbance, melanosis coli and severe drug dependence after long-term use of laxatives [15]. For instance, the research of Zhang et al. [16] confirmed the exact effect and advantages of acupuncture in the treatment of functional constipation and by comparing massage with lactulose oral liquid in the treatment of constipation, Gao et al. [17] showed that tuina was more effective and there was no obvious adverse reaction, the research of Wang

et al. [18] confirmed that acupoint sticking therapy had unique advantages in the treatment of functional constipation because of simple operation, safe and non-invasive, satisfactory curative effect, low cost. The research of Bai et al. [19] confirmed the effect of cupping therapy on constipation and Luo et al. [20] confirmed the curative effect of ear-acupressure on constipation through systematic review. However, whether traditional Chinese external therapies are effective or superior to routine treatment measures remain to be clarified.

In order to further evaluate the curative effect of external therapies of traditional Chinese medicine on constipation in patients with CRF and to provide scientific theoretical basis for clinical practice, this study systematically evaluated and analyzed the clinical research literature of external therapies of traditional Chinese medicine on constipation in patients with chronic renal failure.

This meta-analysis was performed in accordance with the recommendations of the Cochrane handbook for systematic reviews of interventions and was reported in compliance with Meta-Analyses (PRISMA) statement guidelines.

## 2. Methods

### 2.1 Data sources and search strategy

We conducted a search of English database of PubMed, EMBASE, the Cochrane Library and the Web of Science and Chinese database of China National Knowledge Infrastructure (CNKI), Wan-Fang database, VIP Database and China Biomedical Literature Database (CBM) which were published between establishment of the database and December 2022 using the search terms *constipation*, *Renal Insufficiency*, *Chronic*, *Renal Dialysis*, *Acupuncture Therapy*, *Moxibustion*, *Acupoint sticking therapy*, *Tuina*, *Massage*, *Cupping Therapy*, *Auricular point sticking*, *Acupuncture*, *Ear*, *Enema*, *Acupoint Catgut-Embedding Therapy*, *placebo and RCTs*.

### 2.2 Inclusion and exclusion criteria

The inclusion criteria were developed by using a PICOS (Patient, Intervention, Comparators, Outcome, Study design) approach. The study subjects were adult patients who were diagnosed with CRF and constipation and there were specific diagnostic criteria for constipation in the study; the study compared routine treatment combined with different ways of external therapies of traditional Chinese medicine versus routine treatment and there was no limit to the course of treatment; all the studies were randomized controlled trials. The outcome indicators included total effective rate, weekly defecation times, defecation time, clinical constipation scores, PAC-QOL and occurrence of adverse events.

Studies were rejected according to the exclusion criteria as follows: constipation was caused by organic lesions of the intestinal tract(such as tumor, Crohn's disease, colonic polyp, intestinal tuberculosis et al), repeatedly published literature or data repeated in other articles or data included in other articles, no clear evaluation standard of curative effect, no data available for this study in the article, animal experiments, reviews, case reports, expert experience reports, conference abstracts and article not in Chinese or English.

### 2.3 Data extraction

Data from the retrieved literature were extracted and analyzed independently by two reviewers (WY and YQS), if there were different opinions about the quality of a study, a third reviewer examined the controversial literature and discussed it with the two aforementioned reviewers to resolve the discussion. And data were included only if the three reviewers achieved

consensus regarding the data. The extracted data mainly included the first author, publication year, sample size, age, sex, intervention, treatment time, outcome indications and so on.

### 2.4 Quality assessment

Two authors (WY and YYS) independently evaluated risk of methodological quality of the included literature by using the Cochrane risk-of-bias tool. There were seven items: random sequence generation, allocation concealment, blinding of participants and personnel, blinding of outcome assessment, incomplete outcome data, selective reporting, and other bias. The evaluation results of each item were divided into three levels: low-risk, unclear and high-risk, which were represented by a score of bias.

### 2.5 Statistical analyses

Analyses were performed using Review Manager 5.3. Dichotomous outcome data were measured with odds ratio (OR) and 95% CI, and continuous outcomes were measured with mean difference (MD) or standardized mean difference (SMD) and 95% CI, and we assessed changes based on mean values and standard deviations (SDs) changes between the pre-treatment and post-treatment. The heterogeneity between studies was evaluated by the chi-square-based Q statistical test using the heterogeneity $\chi^2$ and $I^2$ statistics. In all analyses, $P < 0.05$ was deemed to represent significant heterogeneity, and the random-effects models were used for the meta-analysis of each indicator and vice versa. In addition, publication bias was assessed by using funnel plots.

## 3. Results

### 3.1 Searching results

The search strategy generated 1409 citations, of which 79 published articles appeared to be relevant to the systematic review and were retrieved for further assessment. Of these studies, 56 were excluded. Overall we included 23 eligible articles reporting on 23 separate trials ultimately. These trails contained 1764 patients who were allocated to test group and control group. The search process was illustrated in Fig 1, and detailed characteristics of the eligible studies were listed in Table 1.

### 3.2 Quality of trials

Of the 23 eligible randomized controlled trials, 18 studies described the grouping method as the random number table method, 4 studies only mentioned "random" and 1 study was numbered according to the order of hospital admission and then grouped according to the random number table method. And only 2 studies were double blinded trials. None of the studies reported the reporting bias and the existence of other biases (Fig 2).

### 3.3 Outcome analysis

**3.3.1 Total efficacy.** 14 studies evaluated the effectiveness of the treatment of which 552 patients were assigned to treatment group and 551 were allocated to control group. 8 studies [21, 23, 24, 27, 28, 30, 32, 40] evaluated the therapeutic effect according to the standard of "Guidance Principle of clinical study on new drug of traditional Chinese medicine [21]". Efficacy means that the time between defecation is shortened by 1 day, and all other symptoms are improved. The failure of treatment means that constipation symptoms do not improve or increase. And 6 studies [22, 25, 31, 33, 35, 37] evaluated the therapeutic effect according to "Standard for traditional Chinese medicine clinical diagnosis and efficacy [37]". The efficacy

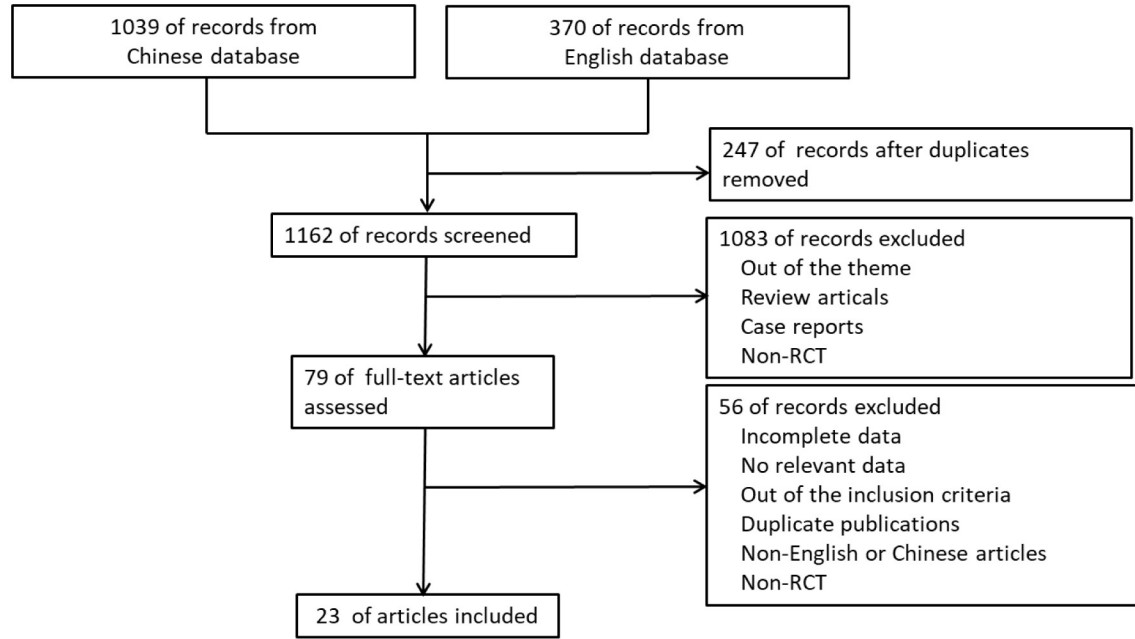

**Fig 1. Flow diagram of literature search and selection of included studies.**

means that the patient can defecate once within 3 days and the clinical symptoms are relieved, the failure of treatment means that constipation symptoms do not improve or increase. Total effective rate = (number of effective cases/total cases) × 100%.

There was no statistical heterogeneity among the studies ($I^2 = 0\%$, P = 0.87), the fixed effect model was used for analysis. The results showed that there was statistical significance in total efficacy between treatment and control group(OR = 4.53, 95%CI(3.27, 6.29), Z = 9.07, P < 0.00001; Fig 3).

**3.3.2 Weekly defecation times.** 7 studies [21, 24, 29, 34, 36, 38, 41] evaluated the weekly defecation times after the treatment of which 266 patients were assigned to treatment group and 265 patients were assigned to control group respectively. There was statistical heterogeneity among the studies ($I^2 = 72\%$, P = 0.002), the random effect model was used for analysis. The results showed that there was statistical significance in weekly defecation times changes between baseline and end of treatment between treatment and control group(MD = 0.94, 95% CI(0.70, 1.18), Z = 7.74, P < 0.00001; Fig 4).

**3.3.3 Defecation time.** 6 studies [21, 24, 29, 30, 38, 41] evaluated the defecation time after the treatment of which 216 patients were assigned to treatment group and 215 patients were assigned to control group respectively. There was no statistical heterogeneity among the studies ($I^2 = 0\%$, P = 0.73), the fixed effect model was used for analysis. The results showed that there was statistical significance in defecation time changes between baseline and end of treatment between treatment and control group(MD = -2.92, 95%CI(-3.69, -2.16), Z = 7.49, P < 0.00001; Fig 5).

**3.3.4 Defecation difficulty score.** 3 studies [26, 31, 35] compared the defecation difficulty score after the treatment of which 115 patients were assigned to treatment group and control group respectively. There was statistical heterogeneity among the studies ($I^2 = 63\%$, P = 0.07), the random effect model was used for analysis. The results showed that was statistical significance in defecation difficulty score changes between baseline and end of treatment between treatment and control group(MD = -1.92, 95%CI(-2.45, -1.39), Z = 7.11, P < 0.00001; Fig 6).

**Table 1. Characteristics of trials included in the study.**

| Study | Sample | Intervention (treatment) | Intervention (control) | Follow-up | Outcomes |
|---|---|---|---|---|---|
| Zhao 2015 [21] | 50(25/25) | Routine treatment + Seed-embedding at onopoints combined with abdominal massage | Routine treatment | 4W | ①②③ |
| Shen 2015 [22] | 90(45/45) | Routine treatment + Auricular-plaster therapy | Routine treatment | 4W | ① |
| Li 2015 [23] | 67(34/33) | Routine treatment + moxibustion therapy | Routine treatment | 8W | ① |
| Zhang 2016 [24] | 60(30/30) | Routine treatment + massage + ultrasonic drug penetration therapy | Routine treatment + massage | 1W | ①②③ |
| Wang 2016 [25] | 100(50/50) | Routine treatment + massage combined with Daihuang Ointment application on Shenque (CV 8) | Routine treatment | 1W | ① |
| Ma 2017 [26] | 50(25/25) | Routine treatment + moxibustion on abdominal acupoints | Routine treatment | 4W | ④ |
| Chen 2018 [27] | 78(39/39) | Routine treatment +Seed-embedding at otopoints combined with abdominal massage | Routine treatment | / | ① |
| Zhang 2018 [28] | 68(34/34) | Routine treatment + auricular pressure | Routine treatment | 8W | ①⑤ |
| Li 2018 [29] | 80(40/40) | Routine treatment +Seed-embedding at otopoints combined with abdominal massage | Routine treatment | 4W | ②③ |
| Li 2018 [30] | 90(45/45) | Routine treatment + Auricular embedding beans combined with abdominal massage | Routine treatment | / | ①③ |
| Huang 2019 [31] | 96(48/48) | Routine treatment +Seed-embedding at otopoints combined with moxibustion therapy | Routine treatment | 1W | ①④ |
| Li 2019 [32] | 60(30/30) | Routine treatment + Acupressure combined with abdominal massage | Routine treatment | 4W | ① |
| Ding 2019 [33] | 60(30/30) | Routine treatment + Umbilical cord steaming therapy | Routine treatment | 4W | ① |
| Abbasi 2019 [34] | 70(35/35) | Routine treatment + Acupressure | Routine treatment | 4W | |
| Lu 2020 [35] | 84(42/42) | Routine treatment +Seed-embedding at otopoints combined with moxibustion therapy | Routine treatment | 1W | ①④ |
| Lou 2020 [36] | 120(60/60) | Routine treatment + Acupoint-sticking therapy | Routine treatment | 4W | ②⑥ |
| Wang 2021 [37] | 120(60/60) | Routine treatment + Seed-embedding at Otopoints combined with acupoint-sticking therapy | Routine treatment | 4W | ①⑥ |
| Zhang 2021 [38] | 68(34/34) | Routine treatment +Seed-embedding at otopoints combined with acupressure | Routine treatment | 4W | ②③ |
| Fu 2021 [39] | 50(25/25) | Routine treatment + Acupoint Embedding at otopoints | Routine treatment | 4W | ⑤ |
| Yan [40] 2021 | 80(40/40) | Routine treatment + Chinese medicine acupoint application | Routine treatment | 4W | ①⑥ |
| Li [41] 2022 | 83(42/41) | Routine treatment + Auricular point pressing combined with Umbilical sticking | Routine treatment | 4W | ②③⑥ |
| Li [42] 2022 | 50(25/25) | Routine treatment + Acupoint Embedding at otopoints | Routine treatment | 48W | ⑤ |
| Chen [43] 2022 | 90(45/45) | Routine treatment +Acupuncture therapy + Chinese medicine acupoint application | Routine treatment | 12W | ⑥ |

①total efficiency; ②Weekly defecation times; ③Defecation time; ④Defecation difficulty score; ⑤Adverse events of treatment; ⑥Patient-Assessment of Constipation Quality Of Life(PAC-QOL)

**3.3.5 Patient-Assessment of Constipation Quality Of Life (PAC-QOL).** 5 studies [36, 37, 40, 41, 43] compared the quality of life in patient with constipation after the treatment of which 247 patients were assigned to treatment group and 246 patients were assigned to control group respectively. There was statistical heterogeneity among the studies ($I^2$ = 64%, P = 0.03), the random effect model was used for analysis. The results showed that there was statistical significance in quality of life in patients between baseline and end of treatment between treatment and control group(MD = -7.57, 95%CI(-10.23, -4.91), Z = 5.58, P < 0.00001; Fig 7).

**3.3.6 Adverse events of treatment.** 3 studies [28, 39, 42] reported the occurrence of adverse events during treatment, including intestinal obstruction, hematochezia and allergic

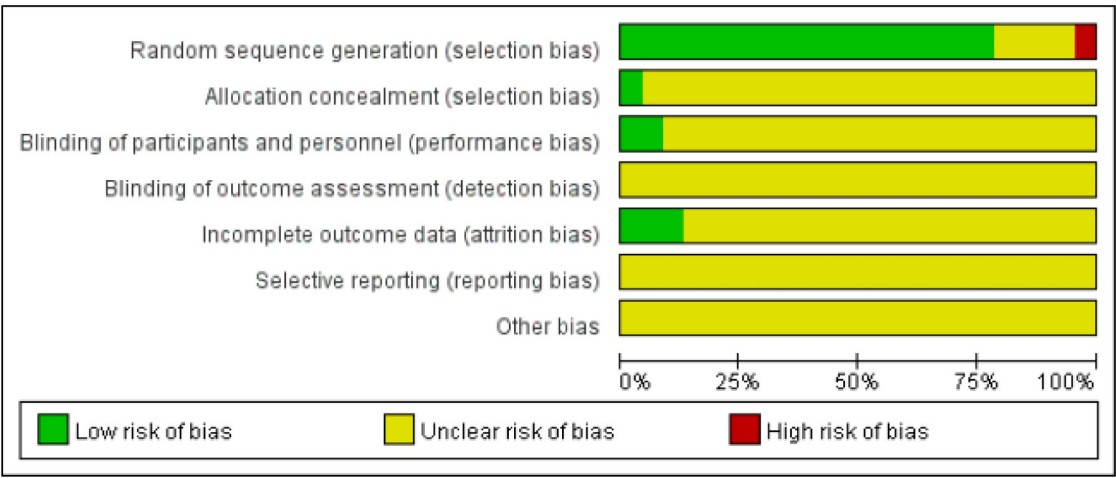

**Fig 2. Risk of bias graph of included studies.**

reactions characterized by skin redness and itching after auricular point sticking. There was statistical heterogeneity among the studies ($I^2$ = 55%, P = 0.11), the random effect model was used for analysis. The results showed that there was no statistical significance in the incidence of adverse events between treatment and control group(OR = 0.35, 95%CI(0.04, 2.95), Z = 0.96, P = 0.34; Fig 8).

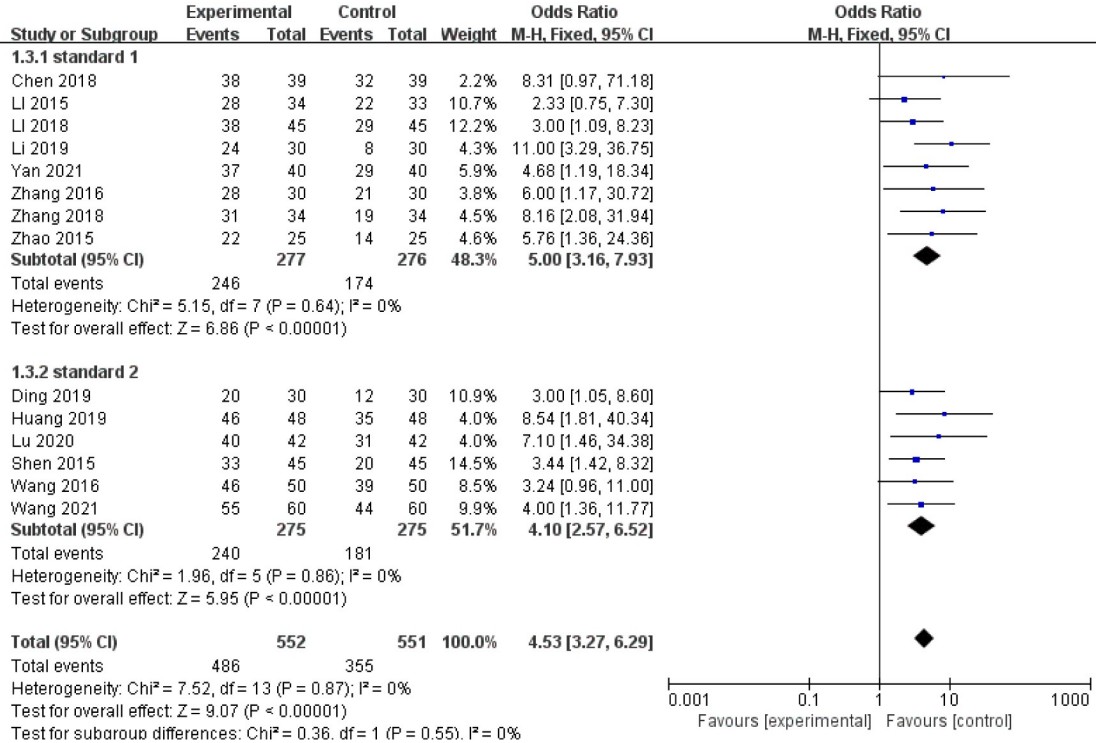

**Fig 3. Forest plot of total efficiency in the treatment group and the control group.**

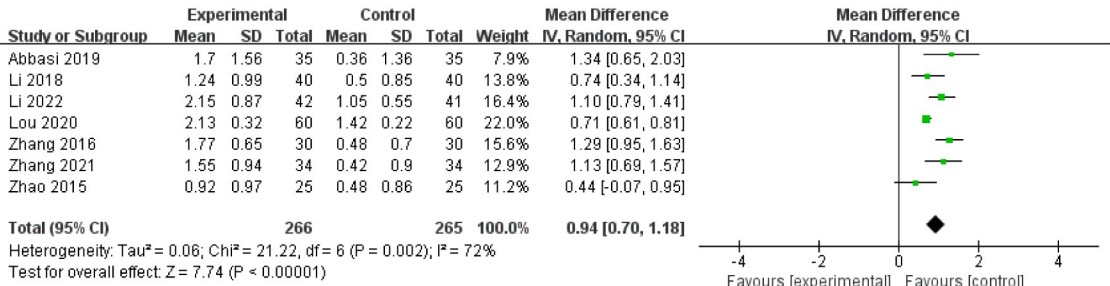

**Fig 4. Forest plot of weekly defecation times changes between baseline and end of treatment between treatment and control group.**

## 3.4 Funnel plot of publication bias

As shown in Fig 9, a funnel chart analysis of clinical effectiveness rate was performed. Results showed that the scatter is basically distributed at the top of the funnel chart which indicated that the risk of publication bias in the clinical efficacy of constipation is low.

## 4. Discussion

Constipation is a disorder with a global prevalence of 14% [2]. It's prevalence increases with age and is almost twice as common in women than men [44], which can significantly interfere with patients' daily life style and sense of well-being and consumes resources in healthcare systems worldwide [45–47]. The prevalence of constipation in dialysis patients is reported to be 36.3%~66.7% [48] and has been shown to be higher in patients with CKD than the general population, particularly among those undergoing dialysis [49, 50].

Water restriction, water removal by dialysis, inadequate intake of dietary fiber due to potassium restriction and the associated changes in intestinal microflora, lack of exercise, diabetic autonomic nervous system disorder, intake of potassium inhibitors and phosphorus adsorbents may be the reasons why dialysis patients are prone to develop constipation [51, 52]. However, A considerable proportion of constipated people were reported dissatisfied with conventional treatments, because of the lack of effectiveness and the side effects [53–56].

In recent years, several studies had shown that external therapies of traditional Chinese medicine such as acupuncture, moxibustion, massage, cupping therapy, ear acupunctures and so on were effective in the treatment of constipation, and compared with conventional treatments, they were more effective, non-toxic and without side effects, low cost, more convenient

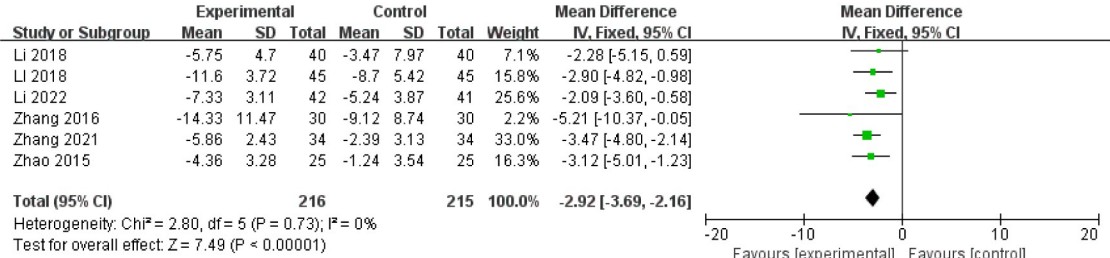

**Fig 5. Forest plot of defecation time changes between baseline and end of treatment in the treatment group and the control group.**

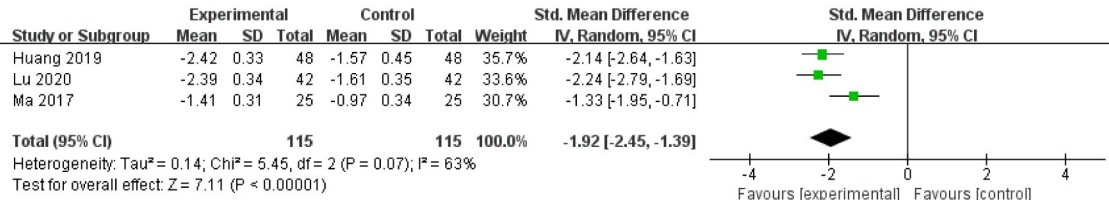

**Fig 6. Forest plot of defecation difficulty score changes between baseline and end of treatment in the treatment group and the control group.**

operation and suitable for long-term medication [14, 57–60]. And in this meta-analysis, the therapeutic effect and safety of external therapy of traditional Chinese medicine on chronic renal failure patients with constipation was explored. 14 studies suggested that the effect of routine treatment combined with different ways of external therapies of traditional Chinese medicine was higher than that of routine treatment, which could improve the efficacy of treatment. Compared with routine treatment group, 7 studies suggested that routine treatment combined with different ways of external therapies of traditional Chinese medicine could increase the weekly defecation times, 6 studies suggested that routine treatment combined with different ways of external therapies of traditional Chinese medicine could shorten defecation time, and 3 studies suggested that routine treatment combined with different ways of external therapies of traditional Chinese medicine could improve the defecation difficulty score and 5 studies reported the life quality of CRF patients with constipation which suggested that routine treatment combined with different ways of external therapies of traditional Chinese medicine could reduce the PAC-QOL score and improve the life quality of patients with chronic renal failure and constipation. At last, 3 the study reported the adverse events occurred in the course of treatment, including intestinal obstruction, hematochezia and allergic

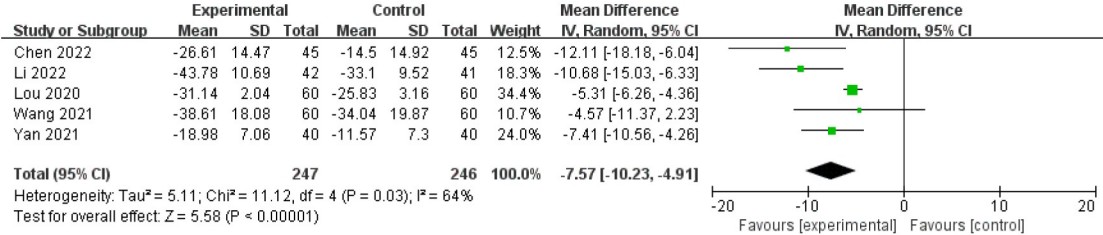

**Fig 7. Forest plot of PAC-QOL changes between baseline and end of treatment in the treatment group and the control group.**

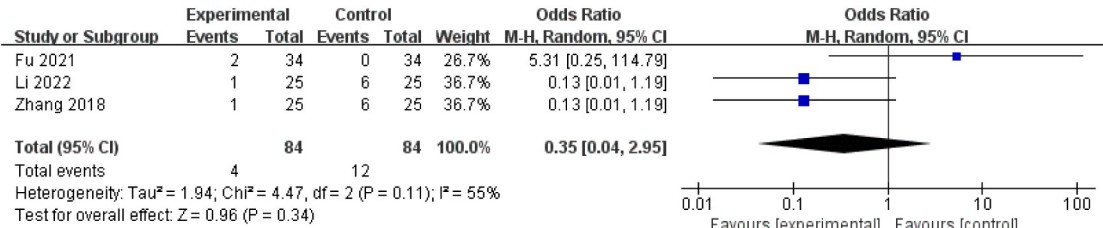

**Fig 8. Forest plot of adverse events in the treatment group and the control group.**

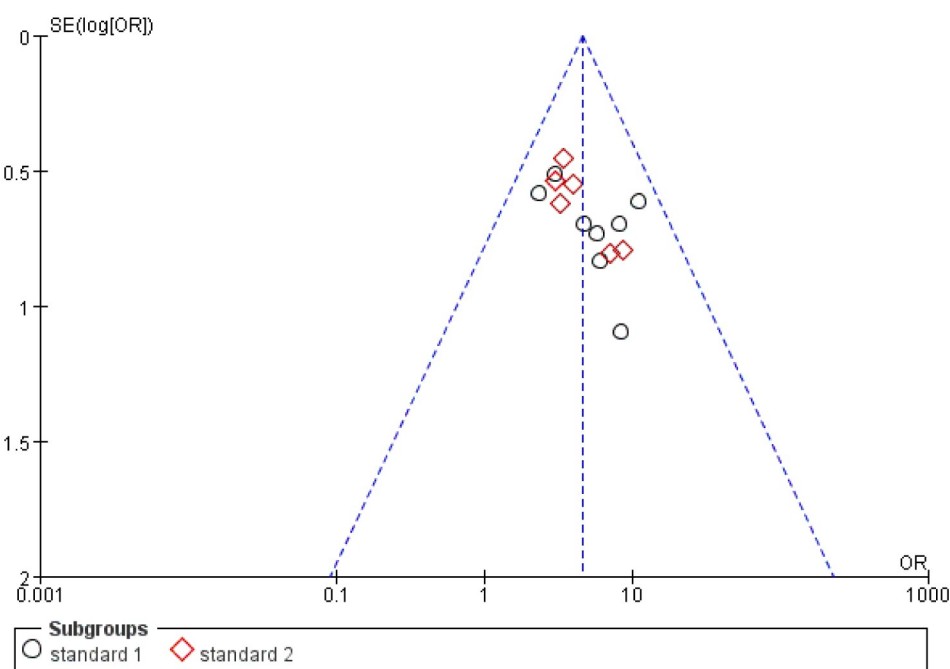

**Fig 9. Funnel plot of effectiveness rate in the treatment group and the control group.**

reactions characterized by skin redness and itching after auricular point sticking, studies suggested that there was no statistical significance in the incidence of adverse events between two groups. We also conducted sensitivity analysis to evaluate the stability of the results of our meta-analysis, and the results suggested that excluding the small sample studies had no significant effect on the results of our meta-analysis.

A meta-analysis [61] in 2022 showed that treatment with Seed-embedding at otopoints plus routine treatment had an better effect on dialysis patients with constipation compared to treatment with routine treatment alone. However, that study only discussed the external therapy of traditional Chinese medicine, which was based on Seed-embedding at otopoints and that study only included patients with maintenance hemodialysis. Our meta-analysis covers a wider range of subjects and treatments.

The meta-analysis had some limitations. First, Some studies did not describe the randomization process or the procedure of allocation concealment in detail; Second, the sample size was small in some studies. Third, most of the studies in this meta-analysis were Chinese-based, which may have caused regional, language, and racial biases. We await further well-designed and high-quality RCTs to further provide reliable data.

## 5. Conclusion

In conclusion, The combination of external therapies of traditional Chinese medicine and routine treatment could achieve an excellent curative effect on constipation in people with CRF. And there were no obvious adverse events in the course of treatment.

## Supporting information

**S1 Data set.**
(XLSX)

**S1 Checklist. PRISMA 2020 checklist.**
(DOCX)

## Author Contributions

**Conceptualization:** Yu Wu, Qisu Ying, Xiangcheng Xie, Xiao Yuan, Ming Wang, Xiao Fei, Xiu Yang.

**Data curation:** Yu Wu, Yajing He.

**Methodology:** Yu Wu, Xiu Yang.

**Supervision:** Xiu Yang.

**Visualization:** Xiangcheng Xie.

**Writing – original draft:** Yu Wu.

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
