## [Decision Letter · Decision Letter 0]

3 May 2023

PONE-D-23-04019Effect of External Therapies of Traditional Chinese Medicine on Constipation in Patients with CRF: A Meta-AnalysisPLOS ONE

Dear Dr. Yu,

Thank you for submitting your manuscript to PLOS ONE. After careful consideration, we feel that it has merit but does not fully meet PLOS ONE’s publication criteria as it currently stands. Therefore, we invite you to submit a revised version of the manuscript that addresses the points raised during the review process.

We look forward to receiving your revised manuscript.

Kind regards,

Tauqeer Hussain Mallhi, Ph.D

Academic Editor

PLOS ONE

“This study was supported by Grants from Zhejiang Provincial Traditional Chinese Medicine Science and Technology Project (2021ZB223).”

Additional Editor Comments:

Dear Authors, thank you for submitting in Plos One. Your manuscript has been assessed by relevant experts from the field. They found the manuscript interesting but raised some concerns in methodology and interpretation of results. It is requested to please consider the comments of reviewers, particularly the need of rationalizing on selection of therapy for this study.

Reviewers' comments:

Reviewer's Responses to Questions

**Comments to the Author**

1. Is the manuscript technically sound, and do the data support the conclusions?

Reviewer #1: Yes

Reviewer #2: Yes

Reviewer #3: Yes

Reviewer #4: Yes

2. Has the statistical analysis been performed appropriately and rigorously? 

Reviewer #1: Yes

Reviewer #2: Yes

Reviewer #3: I Don't Know

Reviewer #4: I Don't Know

3. Have the authors made all data underlying the findings in their manuscript fully available?

Reviewer #1: Yes

Reviewer #2: Yes

Reviewer #3: Yes

Reviewer #4: Yes

4. Is the manuscript presented in an intelligible fashion and written in standard English?

Reviewer #1: Yes

Reviewer #2: Yes

Reviewer #3: No

Reviewer #4: No

5. Review Comments to the Author

Reviewer #1: Thanks to the dear editor and thanks to the researchers who have designed an interesting research. The data source and search strategy were acceptable.

The inclusion and exclusion criteria were designed correctly.

In the results section, due to the wide range of manual methods of Chinese medicine, the conclusion was ambiguous. In fact, it would have been better to remove the methods that had drugs, that is, apart from the manual method, drugs were also used, such as ultrasonic drug penetration ointment or Daihong oil, and for example, three methods such as seed embedding, Acupress, and Acupoint stichy therapy were compared and the conclusions were explained in manual applications. was given Currently, the breadth and variety of methods has caused a lack of clear decision making. In general, considering the prevalence of constipation in kidney failure, it is a valuable work.

Regards

Reviewer #2: The objective of this study was to evaluate the curative effect of external therapies of traditional Chinese medicine in chronic constipation and to provide scientific theoretical basis for clinical practice

This deviates from the title which includes patients with CRF

Besides the study subjects were adult patients who were diagnosed with CRF and constipation.

English language editing required.

Reviewer #3: 1. The grammar for academic writing should be provided in this manuscript.

2. Please clarify the meaning of “CRF” in this work.

3. Introduction

3.1 Please give information about CRF

3.2 The importance of Chinese medicine practices for supportive treatment of constipation should additionally review

4. Methods

4.1 Please give the rationale for selecting Chinese biomedical databases in this work because there are differences in utility and accessibility.

4.2 In search terms, “renal insufficiency” and “Renal dialysis” seem not related to external therapy of conscription. Please clarify (minor issue).

5. Results

5.1 Figure1 there is typing error “recomved”

5.2 Table 1, Three pieces of research with too small sample sizes were analyzed in the work

(Zhao 2015, Fu 2021, Li 2022). The researcher should be concerned that too small a sample size leads to a lack of power in the study. It means that some outcomes have minor clinical interests or effects.

5.3 As reviewer understand in Figure 9 X-axis should be OR [LOG] and Y-axis should be standard error. If reviewer misunderstand, please clarify the results.

6. Discussion and conclusion

6.1 Main point of each outcome indicator after analyzing should be discussed.

6.2 Conclusion demonstrates overclaim results.

Reviewer #4: The article is an interesting one to evolve practice guidelines for the treatment of constipation in CKD through traditional Chinese medicine. The following points may be considered for revision:

1. Grammatical errors as shown may be corrected.

2. How was bias in fig.2 determined and what is other bias?

3. Statistical significance is the right usage term. Kindly correct it accordingly in all places where it is mentioned significantly statistical differences.....

6. PLOS authors have the option to publish the peer review history of their article (what does this mean?). If published, this will include your full peer review and any attached files.

Reviewer #1: No

Reviewer #2: No

Reviewer #3: No

Reviewer #4: No

<quillbot-extension-portal></quillbot-extension-portal>

---

## [Author Response · Author response to Decision Letter 0]

21 Jun 2023

Reviewer’s comments and suggestions

Major issues 

1. Please clarify the meaning of “CRF” in this work. 

Chronic renal failure

2. Introduction 

2.1 Please give information about CRF 

Chronic renal failure (CRF) is the outcome of the continuous progression of various chronic kidney diseases, with metabolite retention and imbalance in the water, electrolytes, and acid base being the main manifestations of CRF 

2.2 The importance of Chinese medicine practices for supportive treatment of constipation should additionally review

There are various ways of external therapies of traditional Chinese medicine of constipation, including acupuncture, moxibustion, massage, cupping therapy and so on. A large number of clinical studies had shown that the external therapies of traditional Chinese medicine was not only effective, but also could it avoid some adverse consequences such as abdominal pain, electrolyte disturbance, melanosis coli and severe drug dependence after long-term use of laxatives. For instance, the research of Zhang etal confirmed the exact effect and advantages of acupuncture in the treatment of functional constipation and by comparing massage with lactulose oral liquid in the treatment of constipation, Gao etal showed that tuina was more effective and there was no obvious adverse reaction, the research of Wang etalconfirmed that acupoint sticking therapy had unique advantages in the treatment of functional constipation because of simple operation, safe and non-invasive, satisfactory curative effect, low cost. The research of Bai etal confirmed the effect of cupping therapy on constipation and Luo etal confirmed the curative effect of ear-acupressure on constipation through systematic review.

3. Methods 

3.1 Please give the rationale for selecting Chinese biomedical databases in this work because there are differences in utility and accessibility.

Chinese Biomedical Literature Database is a comprehensive Chinese medical literature database, which contains more than 1600 Chinese biomedical journals since 1978, as well as the literature records of compilation and conference papers, with an annual increase of more than 400,000 articles and updated every month. CBM is the preferred database for systematic reviewers to retrieve relevant studies in Chinese(Ai C L, Duan Y R, WIFFEN PHIL, etal. Chinese Biomedical Databases: Selection and Search for Trials to Conduct Systematic Reviews [J]. Chinese Journal of Evidence-Based Medicine, 2010,10(06): 749-753.). We can go to the CBM database interface by retrieving http://www.sinomed.ac.cn/.

3.2 In search terms, “renal insufficiency” and “Renal dialysis” seem not related to external therapy of conscription. Please clarify (minor issue).

In this study, we evaluated the curative effect of external therapies of traditional Chinese medicine on constipation in patients with chronic renal failure, so we searched the terms of “renal insufficiency” and “Renal dialysis”.

4. Results

4.1 Figure1 there is typing error “recomved” 

removed

4.2 Table 1, Three pieces of research with too small sample sizes were analyzed in the work(Zhao 2015, Fu 2021, Li 2022). The researcher should be concerned that too small a sample size leads to a lack of power in the study. It means that some outcomes have minor clinical interests or effects.

We conducted sensitivity analysis to evaluate the stability of the results of our meta-analysis, and the results suggested that excluding the small sample studies above had no significant effect on the results of our meta-analysis.

1. Total efficacy. After removing Zhao [14] , the results still showed that there did have statistical significance in total efficacy between treatment and control group〔OR=4.48，95%CI（3.20，6.26），Z=8.75，P < 0.00001〕

2. Weekly defecation times. After removing Zhao [14] , the results still showed that there was statistical significance in weekly defecation times changes between baseline and end of treatment between treatment and control group〔MD = 1.01，95%CI（0.75，1.26），Z=7.68，P < 0.00001〕.

3. Defecation time, After removing Zhao [14] , the results still showed that there was statistical significance in defecation time changes between baseline and end of treatment between treatment and control group〔MD = -3.23，95%CI（-4.24，-2.23），Z=6.32，P < 0.00001〕.

4. Defecation difficulty score. After removing Ma [19] , The results still showed that was statistical significance in defecation difficulty score changes between baseline and end of treatment between treatment and control group〔MD = -2.18，95%CI（-2.56，-1.81），Z=11.5，P < 0.00001〕.

5. Adverse events of treatment After removing Fu [32] and Li[35], The results still showed that there was no statistical significance in the incidence of adverse events between treatment and control group〔OR=0.13，95%CI（0.01，1.19），Z=1.80，P =0.07〕

4.3 As reviewer understand in Figure 9 X-axis should be OR [LOG] and Y-axis should be standard error. If reviewer misunderstand, please clarify the results. 

We used the RevMan software to make the Funnel plot, which used the effect (OR, RR, RD, SMD, etc.) as the X-axis, the reciprocal of the standard error of the effect 1/SE (logOR) as the Y-axis, the X-axis scale with the true number, and the Y-axis scale with SE (logOR). The dots in the chart represented the studies included. The results were more reliable when the sample size was larger and the standard error was smaller, and the dots were concentrated in the narrow area at the top of the funnel plot. The smaller the sample size, the greater the variance, the greater the fluctuation, and the greater the standard error, then the corresponding dots of these studies will be scattered in the wider area of the lower funnel, thus eventually showing an inverted funnel shape. Otherwise the opposite.

5. Discussion and conclusion 

5.1 Main point of each outcome indicator after analyzing should be discussed. 

14 studies suggested that the effect of routine treatment combined with different ways of external therapies of traditional Chinese medicine is higher than that of routine treatment, which can improve the efficacy of treatment. Compared with routine treatment group, 7 studies suggested that routine treatment combined with different ways of external therapies of traditional Chinese medicine could increase the weekly defecation times, 6 studies suggested that routine treatment combined with different ways of external therapies of traditional Chinese medicine could shorten defecation time, and 3 studies suggested that routine treatment combined with different ways of external therapies of traditional Chinese medicine could improve the defecation difficulty score and 5 studies reported the life quality of CRF patients with constipation which suggested that routine treatment combined with different ways of external therapies of traditional Chinese medicine could reduce the PAC-QOL score and improve the life quality of patients with chronic renal failure and constipation. At last, 3 the study reported the adverse events occurred in the course of treatment, including intestinal obstruction, hematochezia and allergic reactions characterized by skin redness and itching after auricular point sticking, studies suggested that there was no statistical significance in the incidence of adverse events between two groups.

5.2 Conclusion demonstrates overclaim results. 

In conclusion, The combination of external therapies of traditional Chinese medicine and routine treatment could achieve an excellent curative effect on constipation in people with CRF. And there were no obvious adverse events in the course of treatment.

---

## [Decision Letter · Decision Letter 1]

18 Jul 2023

PONE-D-23-04019R1Effect of External Therapies of Traditional Chinese Medicine on Constipation in Patients with CRF: A Meta-AnalysisPLOS ONE

Dear Dr. Yu,

Thank you for submitting your manuscript to PLOS ONE. After careful consideration, we feel that it has merit but does not fully meet PLOS ONE’s publication criteria as it currently stands. Therefore, we invite you to submit a revised version of the manuscript that addresses the points raised during the review process.

We look forward to receiving your revised manuscript.

Kind regards,

Keiko Hosohata, Ph.D.

Academic Editor

PLOS ONE

Journal Requirements:

Additional Editor Comments:

To the authors,

Please response to the comments by Reviewer 4.

Reviewers' comments:

Reviewer's Responses to Questions

**Comments to the Author**

1. If the authors have adequately addressed your comments raised in a previous round of review and you feel that this manuscript is now acceptable for publication, you may indicate that here to bypass the “Comments to the Author” section, enter your conflict of interest statement in the “Confidential to Editor” section, and submit your "Accept" recommendation.

Reviewer #1: All comments have been addressed

Reviewer #4: (No Response)

2. Is the manuscript technically sound, and do the data support the conclusions?

Reviewer #1: Yes

Reviewer #4: Partly

3. Has the statistical analysis been performed appropriately and rigorously? 

Reviewer #1: Yes

Reviewer #4: I Don't Know

4. Have the authors made all data underlying the findings in their manuscript fully available?

Reviewer #1: Yes

Reviewer #4: Yes

5. Is the manuscript presented in an intelligible fashion and written in standard English?

Reviewer #1: Yes

Reviewer #4: No

6. Review Comments to the Author

Reviewer #1: Dear authors

Thanks for your attention to the comments. you answered all the comments clearly. and now your article is OK. I hope to see another work from you in the field of acupuncture and chronic kidney disease.

Best regards

Reviewer #4: The following queries or suggestions have not been addressed while responding to reviewers' comments

1. Grammatical errors as shown may be corrected.

2. How was bias in fig.2 determined and what is other bias?

3. Statistical significance is the right usage term. Kindly correct it accordingly in all places where it is mentioned "significantly statistical differences....."

The manuscript with minor editing is uploaded for correction.which should be carried out

7. PLOS authors have the option to publish the peer review history of their article (what does this mean?). If published, this will include your full peer review and any attached files.

Reviewer #1: No

Reviewer #4: No

---

## [Author Response · Author response to Decision Letter 1]

2 Sep 2023

1. Please clarify the meaning of “CRF” in this work. 

Chronic renal failure

2. Introduction 

2.1 Please give information about CRF 

Chronic renal failure (CRF) is the outcome of the continuous progression of various chronic kidney diseases, with metabolite retention and imbalance in the water, electrolytes, and acid base being the main manifestations of CRF 

2.2 The importance of Chinese medicine practices for supportive treatment of constipation should additionally review

There are various ways of external therapies of traditional Chinese medicine of constipation, including acupuncture, moxibustion, massage, cupping therapy and so on. A large number of clinical studies had shown that the external therapies of traditional Chinese medicine was not only effective, but also could it avoid some adverse consequences such as abdominal pain, electrolyte disturbance, melanosis coli and severe drug dependence after long-term use of laxatives. For instance, the research of Zhang etal confirmed the exact effect and advantages of acupuncture in the treatment of functional constipation and by comparing massage with lactulose oral liquid in the treatment of constipation, Gao etal showed that tuina was more effective and there was no obvious adverse reaction, the research of Wang etalconfirmed that acupoint sticking therapy had unique advantages in the treatment of functional constipation because of simple operation, safe and non-invasive, satisfactory curative effect, low cost. The research of Bai etal confirmed the effect of cupping therapy on constipation and Luo etal confirmed the curative effect of ear-acupressure on constipation through systematic review.

3. Methods 

3.1 Please give the rationale for selecting Chinese biomedical databases in this work because there are differences in utility and accessibility.

Chinese Biomedical Literature Database is a comprehensive Chinese medical literature database, which contains more than 1600 Chinese biomedical journals since 1978, as well as the literature records of compilation and conference papers, with an annual increase of more than 400,000 articles and updated every month. CBM is the preferred database for systematic reviewers to retrieve relevant studies in Chinese(Ai C L, Duan Y R, WIFFEN PHIL, etal. Chinese Biomedical Databases: Selection and Search for Trials to Conduct Systematic Reviews [J]. Chinese Journal of Evidence-Based Medicine, 2010,10(06): 749-753.). We can go to the CBM database interface by retrieving http://www.sinomed.ac.cn/.

3.2 In search terms, “renal insufficiency” and “Renal dialysis” seem not related to external therapy of conscription. Please clarify (minor issue).

In this study, we evaluated the curative effect of external therapies of traditional Chinese medicine on constipation in patients with chronic renal failure, so we searched the terms of “renal insufficiency” and “Renal dialysis”.

4. Results

4.1 Figure1 there is typing error “recomved” 

removed

4.2 Table 1, Three pieces of research with too small sample sizes were analyzed in the work(Zhao 2015, Fu 2021, Li 2022). The researcher should be concerned that too small a sample size leads to a lack of power in the study. It means that some outcomes have minor clinical interests or effects.

We conducted sensitivity analysis to evaluate the stability of the results of our meta-analysis, and the results suggested that excluding the small sample studies above had no significant effect on the results of our meta-analysis.

1. Total efficacy. After removing Zhao [14] , the results still showed that there did have statistical significance in total efficacy between treatment and control group〔OR=4.48，95%CI（3.20，6.26），Z=8.75，P < 0.00001〕

2. Weekly defecation times. After removing Zhao [14] , the results still showed that there was statistical significance in weekly defecation times changes between baseline and end of treatment between treatment and control group〔MD = 1.01，95%CI（0.75，1.26），Z=7.68，P < 0.00001〕.

3. Defecation time, After removing Zhao [14] , the results still showed that there was statistical significance in defecation time changes between baseline and end of treatment between treatment and control group〔MD = -3.23，95%CI（-4.24，-2.23），Z=6.32，P < 0.00001〕.

4. Defecation difficulty score. After removing Ma [19] , The results still showed that was statistical significance in defecation difficulty score changes between baseline and end of treatment between treatment and control group〔MD = -2.18，95%CI（-2.56，-1.81），Z=11.5，P < 0.00001〕.

5. Adverse events of treatment After removing Fu [32] and Li[35], The results still showed that there was no statistical significance in the incidence of adverse events between treatment and control group〔OR=0.13，95%CI（0.01，1.19），Z=1.80，P =0.07〕 

4.3 As reviewer understand in Figure 9 X-axis should be OR [LOG] and Y-axis should be standard error. If reviewer misunderstand, please clarify the results. 

We used the RevMan software to make the Funnel plot, which used the effect (OR, RR, RD, SMD, etc.) as the X-axis, the reciprocal of the standard error of the effect 1/SE (logOR) as the Y-axis, the X-axis scale with the true number, and the Y-axis scale with SE (logOR). The dots in the chart represented the studies included. The results were more reliable when the sample size was larger and the standard error was smaller, and the dots were concentrated in the narrow area at the top of the funnel plot. The smaller the sample size, the greater the variance, the greater the fluctuation, and the greater the standard error, then the corresponding dots of these studies will be scattered in the wider area of the lower funnel, thus eventually showing an inverted funnel shape. Otherwise the opposite.

5. Discussion and conclusion 

5.1 Main point of each outcome indicator after analyzing should be discussed. 

14 studies suggested that the effect of routine treatment combined with different ways of external therapies of traditional Chinese medicine is higher than that of routine treatment, which can improve the efficacy of treatment. Compared with routine treatment group, 7 studies suggested that routine treatment combined with different ways of external therapies of traditional Chinese medicine could increase the weekly defecation times, 6 studies suggested that routine treatment combined with different ways of external therapies of traditional Chinese medicine could shorten defecation time, and 3 studies suggested that routine treatment combined with different ways of external therapies of traditional Chinese medicine could improve the defecation difficulty score and 5 studies reported the life quality of CRF patients with constipation which suggested that routine treatment combined with different ways of external therapies of traditional Chinese medicine could reduce the PAC-QOL score and improve the life quality of patients with chronic renal failure and constipation. At last, 3 the study reported the adverse events occurred in the course of treatment, including intestinal obstruction, hematochezia and allergic reactions characterized by skin redness and itching after auricular point sticking, studies suggested that there was no statistical significance in the incidence of adverse events between two groups.

5.2 Conclusion demonstrates overclaim results. 

In conclusion, The combination of external therapies of traditional Chinese medicine and routine treatment could achieve an excellent curative effect on constipation in people with CRF. And there were no obvious adverse events in the course of treatment.

6. How was bias in fig.2 determined and what is other bias?

We assessed risk of bias by using the Cochrane collaboration’s tool[1]. They included selection bias, performance bias, attrition bias, detection bias, reporting bias and other bias.Other bias are other sources of bias which are not addressed in the other domains in the tool.([1]HIGGINS J P, ALTMAN D G, GØTZSCHE P C, etal. The Cochrane Collaboration's tool for assessing risk of bias in randomised trials[J]. BMJ, 2011,343: d5928.DOI: 10.1136/bmj.d5928.)

7.The grammatical errors as shown had been corrected.

---

## [Editor Report · Decision Letter 2]

10 Sep 2023

Effect of External Therapies of Traditional Chinese Medicine on Constipation in Patients with CRF: A Meta-Analysis

PONE-D-23-04019R2

Dear Dr. Yu,

We’re pleased to inform you that your manuscript has been judged scientifically suitable for publication and will be formally accepted for publication once it meets all outstanding technical requirements.

Kind regards,

Keiko Hosohata, Ph.D.

Academic Editor

PLOS ONE

---

## [Editor Report · Acceptance letter]

27 Sep 2023

PONE-D-23-04019R2 

Effect of External Therapies of Traditional Chinese Medicine on Constipation in Patients with CRF: A Meta-Analysis 

Dear Dr. Wu:

I'm pleased to inform you that your manuscript has been deemed suitable for publication in PLOS ONE. Congratulations! Your manuscript is now with our production department. 

Kind regards, 

on behalf of

Dr Keiko Hosohata 

Academic Editor

PLOS ONE